# What Are the Environmental Benefits and Costs of Reducing Food Waste? Bristol as a Case Study in the WASTE FEW Urban Living Lab Project

Eleanor Eaton [1,*], Alistair Hunt [1], Anastasia Di Leo [1], Daniel Black [2,3], Gwen Frost [4] and Sarah Hargreaves [5]

1 Department of Economics, University of Bath, Bath BA2 7AY, UK; a.s.p.hunt@bath.ac.uk (A.H.); adl50@bath.ac.uk (A.D.L.)
2 Daniel Black + Associates | db + a, Bristol BS7 9AZ, UK; daniel.black@bristol.ac.uk
3 Bristol Population Health Sciences, University of Bristol, Bristol BS8 1TH, UK
4 Bristol Waste Company, Bristol BS2 0XS, UK; gwen.frost@bristolwastecompany.co.uk
5 Resource Futures, Bristol BS1 6XN, UK; sarah.hargreaves@resourcefutures.co.uk
* Correspondence: e.a.eaton@bath.ac.uk

**Abstract:** The city of Bristol currently generates around 48,000 tonnes of household food waste every year. This waste incurs loss of resources and environmental damage throughout the food cycle. In this paper we quantify and value the baseline socio-environmental impacts from household food waste in Bristol before examining the potential costs and benefits that may result from changes to food waste behaviour. In so doing, we look to better inform the choice of food waste reduction methods in public policy. The environmental impacts of two possible policy targets are explored: (1) a 20% increase in food waste recycling and (2) an overall decrease in food waste of 20%. Environmental impacts are estimated for 13 different hazards, including Global Warming Potential, Particulate Matter, Human Toxicity and Water Depletion. The societal consequences of these environmental changes are monetised using non-market values which allows us to directly compare the relative importance of different environmental impacts and the trade-offs between these impacts in each scenario. For example, we estimate that the Global Warming Potential of Bristol's annual food waste equates to around 110,000 tonnes $CO_2$, or 25,000 additional cars on the road every year. We find that a 20% improvement in recycling behaviour would lead to an annual reduction of 113 tonnes of $CO_2$ equivalent, whilst a 20% reduction in food waste would result in an annual reduction of 15,000 tonnes $CO_2$ equivalent. Findings suggest that the environmental impact of waste management is significantly overshadowed by the impact of resources used in food production and distribution before it becomes waste.

**Keywords:** food–energy–water nexus; food waste; resource efficiency; non-market valuation; environmental economics; urban living lab; waste management

## 1. Introduction

A recent estimate suggests that about 30% of food production is wasted [1]. In 2018, total wasted food for all sectors in the UK was estimated at more than 9.5 million tonnes, 68% of which was judged to be avoidable. For households alone, excluding other sectors, the total avoidable food waste was 4.5 million tonnes [2]. This constitutes a considerable burden in terms of resource use, and in the environmental impacts of food creation and waste disposal. The co-existence of food waste with food scarcity across populations further highlights system inefficiencies [3]. Informed by this, the UK government is currently committed to a 20% reduction in food waste by 2025 [4].

In this paper we focus on patterns of food waste in the City of Bristol. This geographical focus serves as an example of a local authority that has the responsibility for facilitating and managing waste collection in its jurisdiction. In 2015 Bristol became the UK's first European

Green Capital. The city remains committed to improving its environmental performance and reducing environmental impacts of energy, waste, water and carbon [5]. Food is a key focus of this commitment: Bristol Going for Gold was launched in 2019, joining together individuals, organisations and policymakers behind a shared ambition of making Bristol a Gold Sustainable Food City [6].

The aim of this study is to model, test and quantify the environmental impacts of changes in food waste recycling behaviours in the area of Bristol. We therefore look to provide responses to the two questions:

- What are the non-market and socio-environmental benefits of reduced food waste along the food/waste cycle through increased food waste recycling?
- What reductions in energy and other resource usage in food production/transport and waste disposal might be gained from reducing food waste by 20%?

Figure 1 illustrates the concept behind this study. Wasted food has two main resource flows: as food it embodies all the resources which have gone into its production and supply. When food becomes household waste, it is disposed of in several ways in Bristol. Each waste disposal method has implications for resource use, and also outputs in the form of products such as biogas.

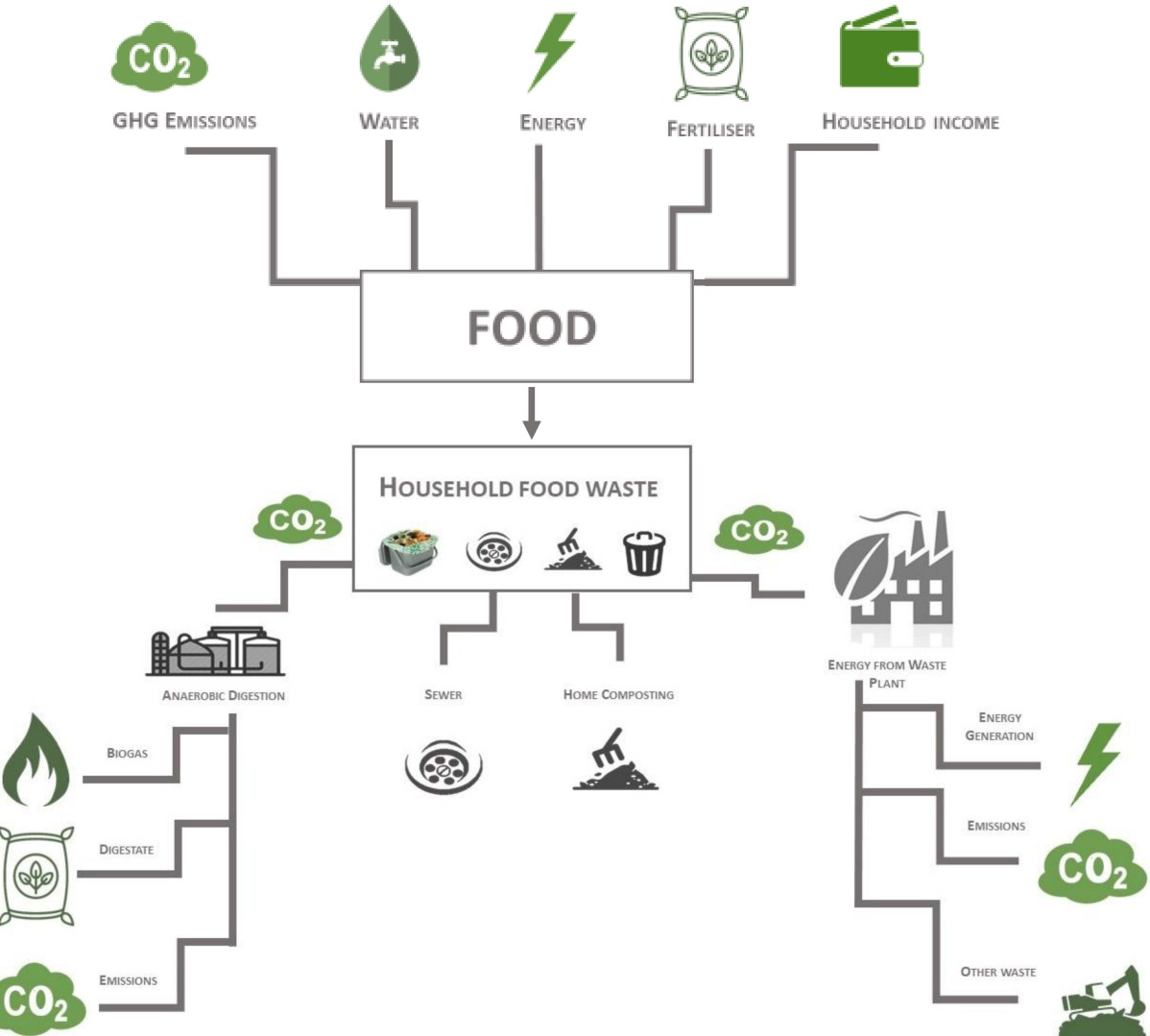

**Figure 1.** Bristol Food Waste Map, illustration of resources going into food and outputs from food waste.

Household food waste refers to food (and drinks) and the inedible parts of food obtained by households, which are not consumed, but instead disposed of via domestic

waste disposal and recycling collections, or alternatively disposed of via home composting or domestic drainage into the sewer network. We exclude food repurposed for animal feed, and food wasted outside the home, such as in restaurants, the workplace or other settings such as schools, hospitals and prisons.

### 1.1. Related Literature

This study focuses on issues relating specifically to household food waste. Households are end users in a whole system of food production and supply, though the household sector is perceived as the largest source of food waste in the UK context [1,7]. However, all elements of the food cycle affect the extent to which households waste food; in particular the role of supermarkets in influencing the availability, longevity and price of food for consumers is recognised [8–10].

Aside from improving efficiencies in the food supply chain, the focus of many policies has been on how to change household behaviour towards reducing unnecessary food purchases and preventing food being wasted within the home [8,11]. Uncertainty remains as to the efficacy of these strategies: it has been shown that the carbon footprint of substitute goods could reduce the stated benefit of household food waste reductions through the rebound effect income savings from reduced food purchases being spent on alternative goods with high embedded carbon—by up to 60% [12].

Mechanisms to change household behaviour are also problematic: for example, the relative low cost of food in high income countries may drive high levels of food waste, but methods to increase the price of food may lead to inequity and increased food insecurity [1,13]. There is a significant lack of data on the efficacy of alternatives focusing on consumers, such as educational programmes, cooking classes, advertising campaigns and food sharing apps [8]. Therefore, the challenge for policy makers is not only to understand the scale of food waste impacts, but also how to appraise alternative reduction strategies.

The environmental impacts of food waste across the whole food cycle have been quantified for the UK by Tonini et al. [7] in detail for 10 different categories of environmental impact. The findings from this study are applied in our own work to estimate the burden at the city level. Taking a similar whole food cycle approach, Chapagain and James explored the water and carbon footprint of avoidable food waste in the UK [14]. Both the Tonini et al. and the Chapagain and James studies used bottom-up Life Cycle Analysis (LCA) methods to estimate all attributable resources used in the production and disposal of food. One alternative to these is the Input–Output (IO) method, using a top-down approach, which takes into account sectoral components and therefore brings in economic estimations. A process-specific LCA starts by compiling data from individual processes within the system boundary. However accurate these are, process-specific LCA is based on an incomplete system, since not all inputs and outputs may be covered by the process-based system [15]. In contrast, the merit of national input–output tables is that they fully cover economic activities within national borders, so that the system is relatively complete. However, the completeness in terms of a system boundary is acquired at the cost of poor resolution in terms of industry classification, as well as the loss of process specificity [16].

Taking a hybrid approach, where the level of detail of a process-based LCA is combined with the completeness of IO models, is suggested by many as the way forward to mitigate uncertainty in compiling life cycle inventories. Some studies use a hybrid model to consider environmental and social impacts of food waste interventions alongside the macro-economy of the agri-food sector in Europe, Australia and Ireland [17–19]. These hybrid studies are useful but limited in the range of environmental impacts considered—whereas Tonini et al. [7] gives 10 environmental categories, these studies explore only three or four categories. Environmental benefits resulting from reductions in food waste are presented against reductions in the agri-food economy with implications for jobs and incomes. However, the value of environmental impacts is not estimated, and therefore comparisons can only give limited insight into the trade-offs between environmental and economic effects.

The choice of method for managing food waste disposal is also key to reducing its environmental impact. Waste processing brings with it both opportunities and costs, in terms of the resources involved in collection, transportation and processing of waste, alongside the economic and environmental benefits of using that waste stream to recover resources and generate alternative products.

Several studies explore alternatives for management of municipal waste in the UK, including food waste, by looking at specific case studies for the London Borough of Greenwich, the London Olympic Park site and Cardiff [20–22]. Environmental endpoints tend to be limited to four or five impacts, but a national study, by Slorach et al., is more comprehensive; specifically comparing household food waste disposal via Anaerobic Digestion (AD) with incineration and landfill for 19 environmental categories in the UK [23].

In these studies, the environmental costs of waste disposal are counterposed against the benefits arising from end-products such as biogas for energy generation, represented by displaced existing grid energy reliant on fossil fuels and other finite resources. Those methods which encourage recycling and optimise the generation of biogas, especially AD, are identified as preferred options. Studies emphasise the importance of local variables such as availability of waste collection methods, waste composition and technical variations within individual waste management types. However, food waste is treated from the point of collection so minimisation is not considered. An exception to this is an Irish study which explicitly compared food waste minimisation with utilisation strategies and warned that the benefits of AD should not be allowed to create demand which incentivises food waste [19].

For the policy maker at the city level, the existing literature does not provide enough detail to model the trade-offs for each policy option tailored to the local context. Our study is unusual in that it attempts to advance a methodology for estimation of the magnitude of impact of improved food waste recycling with interventions to minimise food waste generation at the municipal level. Inspired by the food–energy–water nexus approach, we consider the food cycle in totality, using monetary valuation of environmental impacts to quantify the scale of these effects [24].

Monetary valuation of the environmental impact of waste disposal is rare: WRAP estimated the "true cost" of food waste in the national retail and hospitality sectors, incorporating expenditure relating to food purchases, energy, water, transport and waste management, but did not consider non-market environmental impacts in their calculations [25]. Valuation, where it is used, tends to be employed as a means of defining the scale of environmental externalities in aggregate terms: for example, the FAO has attempted to value the "full cost" of food wastage footprint on a global scale [26]. It has been noted that in some contexts the presentation of true cost accounting may be unhelpful or counterproductive for stakeholders, as it may be complex to implement and communicate [27]. However, we explore how valuation of externalities can support decision making by enabling the comparison of different intervention outcomes, and therefore providing information for effective economic appraisal of food waste management options.

### 1.2. What This Paper Adds

This study seeks to address gaps in the present literature by:

- Specifically advancing a methodology which allows the policy maker to quantify environmental impacts at the city scale.
- Incorporating a whole food life cycle approach to compare options for food waste minimisation against food waste management optimisation.
- Using valuation of environmental externalities to compare scenarios and inform policy appraisal.

### 1.3. Structure of the Paper

The remainder of this paper is structured in the following way:

Section 2 outlines the methodological approach to quantification of baseline and policy scenarios at the city scale, including the key data sources which inform this study. Section 3 presents the results of estimations for the baseline and compares the baseline with each policy scenario. This is done in three stages: baseline impacts for resources which go into food before it is wasted are quantified; waste disposal effects are identified; and finally results are considered for the whole food cycle, from food production through to waste management. Section 4 then provides a discussion on the results, including the limitations of the study and comparisons with existing literature, before ending with some final conclusions on the value of our research.

## 2. Materials and Methods

Our approach was made up of five main steps:

1. Baseline Production. Define the baseline production of household food waste for the City of Bristol in the year ending March 2019, separated into avoidable and non-avoidable food waste. We follow here the definitions of "avoidable" and "unavoidable" food waste as used by Tonini et al. [7]; avoidable refers to that part of food which prior to being disposed of was edible at some point. Unavoidable food waste refers to the components of food waste which could not be consumed, such as bones, eggshells, etc.

2. Scenario Development. Model two scenarios:

   a. A 20% increase in food waste recycling.
   b. A 20% reduction in food waste, from, e.g., improved food preparation methods.

3. Measurement of Environmental Impact. Calculate the baseline environmental impacts at current levels of food waste and model the changes which the different scenarios have at different levels of reduction.

4. Valuation of Environmental Impact. Derive values per tonne of household food waste, estimated by the monetary valuation of the environmental impacts of resources which are used in the production and supply of food to households and food preparation methods—"pre-loaded resources".

5. Calculate the costs and benefits of different scenarios.

Consultation was carried out with industry stakeholders Bristol Waste and Wessex Water, who together are responsible for much of the domestic waste processing in Bristol, and with local and national experts: Bristol Food Network, Resource Futures, Centre for Sustainable Energy and the Schumacher Institute. This enabled us to understand and map the local resource flows and refine the model of environmental impacts resulting from food waste.

### 2.1. Baseline Production

Recycled food waste amounts are based on data on food waste placed in recycling bins or caddies in the Bristol area recorded by the WasteDataFlow database [28]. Quantities of food placed in black bins, known as "residual waste", for Bristol were provided by Bristol Waste Company [29], derived from quantities of waste for 2018–2019 from the WasteDataFlow database. Food waste as a proportion of residual waste is calculated on the basis of a 2019 Waste Compositional Analysis Report for Bristol City Council [30].

We then make an additional calculation to derive the amount of waste disposed of via sewers and composted at home, based on national WRAP estimates [31].

We arrive at a total of 47,972 tonnes of food waste per annum for the 200,284 households whose waste is collected by Bristol Waste Company [29]. This equates to around 240 kilos per year per household, or 4.61 kilos avoidable waste per week per household. Calculations may exclude a small number of Bristol residents who do not have their waste collected by Bristol Waste Company, for example, residents of some types of flats.

For collected waste (recycled and residual), it is assumed that the percentage of food which is avoidable is the same as in the Waste Composition Analysis for Bristol [30]. For

other waste, such as that disposed of via home composting and sewer, the proportion of avoidable to unavoidable reflects that in Gillick and Quested [31]. Quantities of food waste separated by disposal method are set out in Table 1.

All waste collected from Bristol household recycled food caddies is sent to an Anaerobic Digestion (AD) plant managed by GENeco in Avonmouth, just outside Bristol [32]. The plant generates digestate, which is used for fertiliser, and biogas, used for energy generation. All residual waste (residual waste is defined as waste not put into recycling; in Bristol household waste not separated for recycling is disposed of via black "wheely" bins) from households is sent to an energy from waste plant, which incinerates the waste and generates energy. Following consultation with Bristol Waste, we assume that no domestic food waste is sent to landfill in Bristol [33]. Quantities of food disposed of informally, such as home composting and via the sewer, are estimated based on data from the Waste and Resources Action Programme [34].

**Table 1.** Quantities and Disposal Method of Household Food Waste in Bristol (2018/2019).

|  | % | Tonnes per Year |
|---|---|---|
| Recycled via caddy—avoidable | 58% | 7868 |
| Recycled via caddy—non-avoidable | 42% | 5792 |
| Food waste in residual—avoidable | 83% | 16,194 |
| Food waste in residual—non-avoidable | 17% | 3247 |
| Sewer—avoidable | 70% | 7773 |
| Sewer—non-avoidable | 30% | 3331 |
| Composting—avoidable | 70% | 2487 |
| Composting—non-avoidable | 30% | 1066 |
| Other—avoidable | 70% | 149 |
| Other—non-avoidable | 30% | 64 |
| Total | **100%** | **47,972** |

### 2.2. Scenario Development

Following consultation with Bristol ULL partners, two scenarios for reducing food waste were modelled in terms of their environmental impacts. For each scenario, a 20% change was modelled, reflecting the Cortauld Commitment to a 20% reduction in food waste [4].

### 2.2.1. Scenario 1: More Food Waste Recycling (20% Increase)

We model changes in recycling behaviour by analysing the effects of increases in food waste recycled, and associated reductions in total food in residual waste. There is no change in the total quantity of food being purchased or waste being produced. In this scenario, we are seeking to identify the environmental impacts that might result if the quantity of food waste usually sent to residual waste processing is sent to recycling instead. This scenario is based on that adopted by the Bristol Waste Company "Slim my Waste" campaign in Bristol started in 2017 to encourage households to use food recycling bins rather than disposing of waste in the residual (black) bins [35].

### 2.2.2. Scenario 2: Reduction in Food Waste (20% Decrease)

This scenario simulates a reduction in the total quantity of food waste as a result of more efficient food consumption within the household, leading to a reduction in the overall amount of food purchased by a household. This is designed to mimic national campaigns such as "Love Food Hate Waste" [11]. The proportion of food sent to each disposal method remains unchanged from the baseline.

All scenarios test for a single, one-off, permanent change, with the assumption that there are no other changes in behaviour. As a consequence, in Scenario 1 the quantity of food waste that is not collected by the local authority (i.e., disposal by home composting

and sewer) is unchanged. In Scenario 2, however, these amounts reduce proportionally to the whole.

### 2.3. Measurement of Environmental Impact

In order to understand the environmental impacts of food waste, we use data derived from life cycle analysis. This is because this method provides the level of granularity in detail required to identify a full range of environmental impacts, to tailor information to the city-level scale and to understand how resource use might change under different scenario conditions [16].

The main data source for the measurement of environmental impact is Tonini et al. [7], which employs a bottom-up Life Cycle Analysis (LCA) method to quantify the environmental impacts of avoidable food waste across the food cycle in the UK. Four components of the food cycle are observed: Processing, Wholesale/Retail, Food Service and Households.

The study calculates the impact throughout the food cycle on 10 categories of potential change. This includes changes in land use for different types of food production, waste during farming and processing, wholesale and retail waste, food purchasing, meal preparation and waste management.

In the Tonini et al. study, the environmental impacts of waste disposal are calculated on the basis of the split between waste disposal processes currently existing at the UK level [7]. However, the UK split between waste disposal methods contrasts with that for Bristol, especially in relation to the proportion of waste processed by anaerobic digestion and landfill, and to a lesser extent incineration. In particular, Bristol has more waste processed by anaerobic digestion than the UK and is unusual in that no food waste goes to landfill—the most costly method in terms of environmental impact. This is illustrated in Table 2 below:

**Table 2.** Comparison of Household Food Waste Processing Methods: Bristol and UK compared.

| Waste Processing Method | UK * | Bristol ** |
|---|---|---|
| Anaerobic Digestion (AD) | 8% | 29% |
| Composting | 8% | 7% |
| Incineration | 33% | 41% |
| Sewer | 23% | 23% |
| Landfill | 28% | 0% |
| Other | 0% | 0% |
| TOTAL | 100% | 100% |

\* UK source: Tonini et al [7], adapted with permission from 2018, *Waste Manag.* (rounded). ** Our calculation for Bristol.

Results are given by Tonini et al. in terms of total impact for different stages of the food cycle. However, these are not disaggregated into impacts for different waste disposal methods. To tailor our findings to the Bristol context and so allow us to better understand the effects of changes to different waste management techniques, a secondary method was applied, outputs of which are summarised in Table 3.

Given that $T$ = total burden of life cycle of wasted food, we adjust the waste disposal element only such that

$$T\ (adjusted) = T - WD_{UK} + WD_{Bristol}$$

where

$T\ (adjusted)$ = total burden of life cycle of wasted food, tailored to Bristol context.
$WD_{UK}$ = Net impact attributed to waste disposal methods tailored to UK specific practices.
$WD_{Bristol}$ = Net impact attributed to waste disposal methods tailored to Bristol specific practices.

**Table 3.** Adjustment of LCA impacts for Bristol waste disposal methods.

| | Unit | $T$ (Base) | $WD_{UK}$ | $T - WD_{UK}$ | $WD_{Bristol}$ | $T$ (Adjusted) |
|---|---|---|---|---|---|---|
| AC Acidification * | Mol H+ | 29.00 | −0.41 | 29.00 | - | 29.00 |
| ECO Ecotoxicity * | $CTU_e$ | 3546.00 | −31.00 | 3546.00 | - | 3546.00 |
| FE Freshwater Eutrophication Phosphorous | kg P-eq. | 0.33 | 0.00 | 0.33 | 0.00 | 0.33 |
| FRD Fossil Resource Depletion | MJ | 16,824.00 | −2120.00 | 18,944.00 | −551.79 | 18,392.21 |
| GWP Global Warming Potential | kg $CO_2$-eq. | 2413.00 | 111.00 | 2302.00 | −21.97 | 2280.03 |
| HT Human Toxicity ** | kg 1,4-DB eq. | - | - | - | −8.79 | −8.79 |
| HT Human Toxicity Cancer | $CTU_h$ | $3.36 \times 10^{-5}$ | $-2.40 \times 10^{-6}$ | $3.60 \times 10^{-5}$ | - | $3.60 \times 10^{-5}$ |
| ME Marine Eutrophication Nitrogen | kg N-eq. | 13.00 | 0.36 | 12.64 | 0.35 | 12.99 |
| PED Primary Energy Demand $ | Gj | - | - | - | −1.37 | −1.37 |
| PM10 Particulate Matter ** | kg $PM_{10}$-eq. | - | - | - | 0.51 | 0.51 |
| PM2.5 Particulate Matter | kg $PM_{2.5}$-eq. | 1.95 | −0.05 | 2.00 | - | 2.00 |
| POF Photochemical Ozone Formation | kg NMVOC-eq. | 7.00 | 0.25 | 6.75 | 0.56 | 7.31 |
| WD Water Depletion | $m^3$ water | 3.81 | −0.36 | 4.17 | −180.87 | −176.71 |

(Functional unit: 1 tonne of household food waste); $T$ (Base): total LCA impact (Tonini et al. [7], adapted with permission from 2018, *Waste Manag.*); $WD_{UK}$: net impact of $WD_{UK}$ only (Tonini et al. [7]); $T - WD_{UK}$: net impact of food waste up to point of disposal; $WD_{Bristol}$: net impact of $WD_{Bristol}$ (Slorach et al. [17], adapted with permission from 2019, *Resour. Conserv. Recycl.*); $T$ (Adjusted): total LCA impact adjusted for Bristol WD methods; * not covered by Slorach et al., so UK impact unadjusted; ** alternative metric used by Slorach et al.; $ not covered by Tonini et al.

To derive $T - WD_{UK}$ we refer to Tonini et al.'s LCA results to remove the effects of waste management ($WD_{UK}$) from the household results. To derive $WD_{Bristol}$ we estimate the impacts which different food waste disposal methods have on the environment, based on findings from Slorach et al. [17]. The authors compare the impact of 1 tonne of food waste disposed of by Anaerobic Digestion (AD) (base case) with two other methods: incineration with energy recovery and landfill with gas utilisation. The study also compares the generation of electricity by AD with other electricity generation methods, using a life cycle assessment approach to evaluate the total impact which each method has from the point of collection from the household. Data are drawn from UK AD plants.

The Slorach et al. study does not consider the environmental impact of informal waste management routes, such as home composting and disposal via the sewer. However, it is useful in disaggregating the impact of each principal individual waste disposal method.

Thirteen out of 19 categories of environmental change associated with Anaerobic Digestion (AD) are found by Slorach et al. to have environmental improvements, which they attribute to the displacement of grid electricity and reduction in mineral fertilisers. Incineration is indicated to have negative values in 8 categories, with some environmental benefits also relating to displacement of grid energy. Landfill is the worst alternative compared to incineration or AD.

The combination of two studies in this way was not without its difficulties: two units, Human Toxicity and Particulate Matter, are not equivalent in the two studies and have had to be separately identified. Similarly, Slorach et al. do not approach acidification and ecotoxicity in the same way as the primary study, so we could not adjust these values for the Bristol context. Finally, there may be problems around the definitions used in each study for water depletion, which affect the robustness of results.

*2.4. Valuation of Environmental Impact*

A key aim of this study was to quantify in monetary terms the value of the environmental impact of changes to resource availability and pollution as a result of reducing food waste. Non-market valuation techniques are used to estimate the economic value of goods or services which cannot be bought or sold in a normal market. The effects on human welfare that may be valued using non-market valuation methods may include—for example—the disutility of the impact on human health, the ecosystem and loss of habitat quality and the loss of visual amenity which pollution can have in the environment.

Non-market values are often derived by measuring an affected population's explicit or implicit willingness to pay to avoid a loss of amenity or welfare. However, values

derived in this way can be heavily dependent on factors such as local population income and socio-economic status, the affected population's reliance on natural resources and their experience of pollution events, so there are wide ranges of uncertainty that need to be factored in when used in decision making.

Economic valuations for the social and environmental impact of food waste are therefore intended not to be precise forecasts of an impact on a specific population, but to provide a broad indication of the relative scale of impact by one environmental factor compared to another.

Given the scope of this study, it is not feasible to attempt to quantify the economic value of individual health endpoint changes related to each change in resource. Therefore, we use evidence which has already been identified in the existing published literature for the impacts identified in our study. For example, we are using evidence which provides values for the damage cost of $CO_2$ as a proxy for Global Warming Potential, rather than identifying every impact which the forecast change in $CO_2$ levels may have on the population of Bristol. We refer to these environmental proxy impacts as midpoints. We summarise the unit values for each impact midpoint in Table 4 below. Data on unit costs have been derived from several sources, as there is variation between methodologies and types of environmental pollutant equivalents in the economic valuation literature.

**Table 4.** Summary of midpoint unit values used in the study.

| Impact | Source | Unit | Unit Value | Low | High |
|---|---|---|---|---|---|
| AC Acidification | Trinomics (2020) [36] | £/Mol H+ | 0.29 | 0.15 | 1.39 |
| ECO Ecotoxicity | Trinomics (2020) [36] | £/CTU$_e$ | $3.27 \times 10^{-5}$ | $2.05 \times 10^{-24}$ | $1.61 \times 10^{-4}$ |
| FD Fossil Resource Depletion | Trinomics (2020) [36] | £/MJ | $1.11 \times 10^{-3}$ | $1.11 \times 10^{-3}$ | $5.83 \times 10^{-3}$ |
| FE Freshwater Eutrophication | De Bruyen (2018) [37] | £/kg P-eq. | 1.59 | 0.52 | 34.29 |
| GWP Global Warming Potential | De Bruyen (2018) [37] | £/kg $CO_2$-eq. | 0.05 | 0.02 | 0.09 |
| HT Human Toxicity | De Bruyen (2018) [37] | £/kg 1,4 DB-eq. | 0.08 | 0.08 | 0.08 |
| HT Human Toxicity Cancer | UPSTREAM (2019) [38] | Per case cancer | 32,487 | 31,425 | 975,505 |
| ME Marine Eutrophication | De Bruyen (2018) [37] | £/kg N | 2.67 | 2.67 | 2.67 |
| PED Primary Energy Demand | De Bruyen (2018) [37] | £/Kwh | 25 | 25 | 25 |
| PMF Particulate matter formation | De Bruyen (2018) [37] | £/kg $PM_{10}$-eq. | 22.80 | 19 | 41 |
| PMF Particulate Matter Formation | De Bruyen (2018) [37] | £/kg $PM_{2.5}$-eq. | 33.18 | 27.7 | 59.5 |
| POF Photochemical Ozone Formation | De Bruyen (2018) [37] | £/kg NMVOC-eq. | 0.99 | 0.99 | 0.99 |
| Value of Food Waste (avoidable only) | WRAP (2020) [2] | £/tonne | 2850 | 2850 | 2850 |
| WD Water Depletion | Nematchoua (2019) [39] | £/m$^3$ | 0.07 | 0.07 | 0.07 |

Values converted to £2019.

Values for environmental prices are derived from studies that use a life cycle assessment approach, incorporating values for damage costs of a range of pollutants and environmental midpoints which incorporate human health, ecosystem impacts and resource availability. Environmental impacts are considered at the global scale, reflecting the global origins of food.

## 3. Results

### 3.1. Quantities and Disposal Method of Food Waste

As outlined in Figure 2 and Table 5, we estimate that Bristol households produced around 48,000 tonnes of food waste in the 2018–2019 financial year (April 2018 to March 2019), of which 33,101 are collected at the doorstep.

Around 70% of food waste is estimated to be avoidable, but we find that there is some variation between residual and recycled material. Eighty-three percent of food waste in residual bins is estimated to be avoidable, whereas only 58% of material in food recycling caddies is avoidable [30].

We have assumed these proportions remain the same when modelling for each scenario, such that more recycling leads to less avoidable waste in recycling caddies, and reductions in waste lead to reductions in both avoidable and unavoidable food waste.

Table 5 shows the effect of the assumption that in Scenario 1 food waste (FW) recycled by caddy increases by the same amount that avoidable food waste in residual waste falls.

In Scenario 2 food waste recycled by caddy falls by the same proportion as all other FW components.

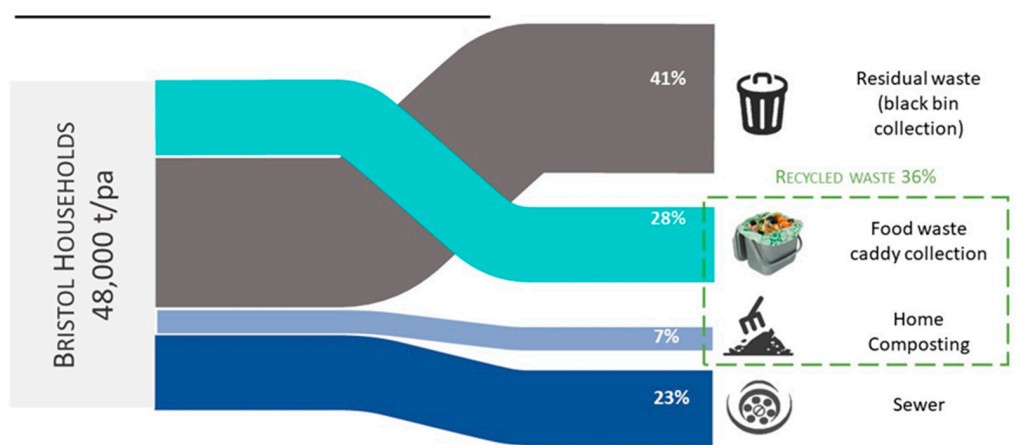

**Figure 2.** Illustration of quantities of household food waste and disposal methods for Bristol.

**Table 5.** Quantities and disposal method for Bristol household food waste (tonnes per annum) under each scenario at 20%.

| Collection Method | Scenario 1 (20%) Change in Recycling | | | | | Scenario 2: (20%) Reduction in Waste | | | | |
|---|---|---|---|---|---|---|---|---|---|---|
| | Avoidable | Non-Avoidable | Total | % | Change | Avoidable | Non-Avoidable | Total | % | Change |
| Recycled (Caddy) | 10,108 | 7440 | 17,548 | 37% | 3888 | 6295 | 4634 | 10,928 | 28% | −2732 |
| Residual | 12,955 | 2597 | 15,552 | 32% | −3888 | 12,955 | 2597 | 15,552 | 41% | −3888 |
| Sewer | 7773 | 3331 | 11,105 | 23% | - | 6219 | 2665 | 8884 | 23% | −2221 |
| Home Composted | 2487 | 1066 | 3553 | 7% | - | 1990 | 853 | 2843 | 7% | −711 |
| Other | 149 | 64 | 213 | 0% | - | 119 | 51 | 171 | 0% | −43 |
| Collected at Doorstep | 23,063 | 10,038 | 33,101 | 69% | - | 19,250 | 7231 | 26,480 | 69% | −6620 |
| **All Food Waste** | **33,473** | **14,499** | **47,972** | **100%** | **-** | **27,578** | **10,800** | **38,377** | **100%** | **−9594** |

Collected at doorstep = Recycled + Residual.

Both scenarios lead to a reduction of 3888 tonnes of food waste from residual bins. A 20% change in recycling (Scenario 1) relocates these 3888 tonnes of food waste to recycling; a small proportion of the overall burden of household waste moving from one form of waste management to another.

Alternatively, a 20% change in the generation of food waste (Scenario 2) affects the entire burden of food waste, leading to a reduction of 9594 tonnes of food waste.

In the next sections we explore the socio-environmental impact of existing levels of food waste, and how these might change under each scenario.

### 3.2. The Impact of Resources Pre-Loaded into Food before It Becomes Waste

A large number of resources go into food before it leaves the household as food waste. These resources relate to the global nature of Bristol's food supply chains, and impacts are likely to be felt worldwide. The creation of food includes resources which reduce air, water and soil quality, and damage human health, as well as reducing primary resources such as water and fossil fuels.

The resources embodied in Bristol food waste are estimated to include damages equivalent to 110,000 tonnes $CO_2$, or the equivalent of 25,000 additional cars on the road every year (Table 6).

We have included in this table the total estimated quantity of food waste in Bristol, including that disposed of via sewers and home composting, in order to indicate the overall impact which a 20% reduction may have on total waste.

Avoidable collected waste forms around half of total waste, which means that the effects of changes to recycling behaviour relate to only a small proportion of total waste, whereas changes to food waste at source are assumed to affect the whole of the total food waste burden.

The zero changes in the total food waste column under Scenario 1 reflect that there is no reduction in food waste overall. Some minor changes in avoidable collected waste are indicated under Scenario 1, which relates to the reduction in the proportion of avoidable food waste in recycled waste compared to residual.

Scenario 2 shows a 20% decrease in both total and avoidable collected food waste, reflecting the 20% reduction in food waste at source.

**Table 6.** Comparison of baseline and changes in resource load of food waste (FW) in Bristol per annum for each scenario (rounded).

| | | Baseline | | Scenario 1 (20%) Change in Recycling | | Scenario 2: (20%) Reduced Food Waste | |
|---|---|---|---|---|---|---|---|
| | Unit | Total FW | Avoidable Collected | Total FW | Avoidable Collected | Total FW | Avoidable Collected |
| Global Warming | t $CO_2$-eq. | 110,400 | 55,400 | 0 | −2300 | −22,100 | −11,100 |
| Acidification | Mol H+ × 1000 | 1500 | 700 | 0 | −29 | −280 | −140 |
| Photochemical Ozone Formation | t NMVOC-eq. | 300 | 200 | 0 | −7 | −60 | −30 |
| Particulate Matter | t $PM_{2.5}$-eq | 100 | 50 | 0 | −2 | −19 | −10 |
| Marine Eutrophication | t N-eq. | 600 | 300 | 0 | −13 | −120 | −60 |
| Freshwater Eutrophication | t P-eq | 20 | 10 | 0 | −0.3 | −3 | −2 |
| Human Toxicity * | CTUh | 2 | 1 | 0 | <1 | <1 | <1 |
| Ecotoxicity | CTUe × 1000 | 170,000 | 85,300 | 0 | −3500 | −34,000 | −17,100 |
| Fossil Resource Depletion | GJ | 909,000 | 456,000 | 0 | −18,900 | −181,800 | −91,200 |
| Water Depletion | $m^3$ water | 200,000 | 100,000 | 0 | −4000 | −40,000 | −20,000 |

\* Human Toxicity is measured in toxicity units equivalent to cases of cancer.

Once we estimate the societal value of these environmental endpoints, through application of the unit values presented in Table 4, we can see the scale of these impacts (as well as the value of using a financial value to compare relative weights) in Figure 3 below.

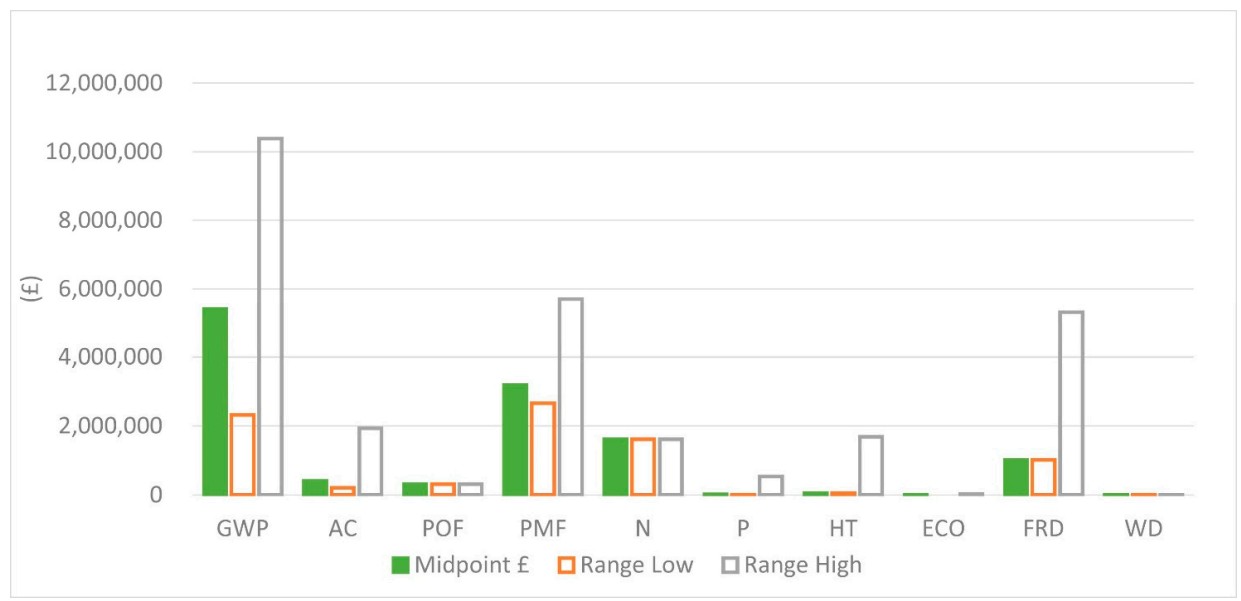

**Figure 3.** Baseline societal value of environmental impacts of resources loaded into food for all Bristol food waste per annum. (Key: GWP: Global Warming; AC: Acidification; POF: Photochemical Ozone Formation; PMF: Particulate Matter Formation; N: Aquatic Eutrophication (Nitrogen), P: Aquatic Eutrophication (Phosphorous); HT: Human Toxicity (Cancer); ECO: Ecotoxicity; WD: Water Depletion).

The ranges in Figure 3 above relate to uncertainties in the economic valuation of unit cost per impact midpoint. In the case of Human Toxicity this variance is very large, as unit costs for cases of cancer vary widely based on the severity of illness.

### 3.3. The Impact of Waste Disposal

This section estimates the impacts of the 33,101 tonnes of food waste collected via household residual bins or recycling caddies, which is directed to either anaerobic digestion or energy from waste processing plants. The environmental impact of informal disposal methods such as sewer or home composting is not detailed in the life cycle analyses used here. This may underestimate the overall impact of total food waste disposal, as we are not able to estimate the environmental impact of food disposed of via these informal routes.

#### 3.3.1. Baseline

Table 7 illustrates the largely beneficial impact which food waste processing has via Anaerobic Digestion (AD) and energy from waste (incineration), in terms of outputs such as energy generation as an alternative to grid electricity and displacement of mineral fertilisers. This table illustrates the marginal impact of waste disposal per tonne of food waste at baseline, and under each scenario.

**Table 7.** Marginal impacts from waste disposal for all Bristol household food waste collected at doorstep (rounded), comparing baseline with changes under each scenario.

| | Unit | Baseline | Scenario 1 (20%) Change in Recycling | Scenario 2 (20%) Reduced Food Waste |
|---|---|---|---|---|
| Primary Energy Demand | GJ | −1.37 | −1.50 | 1.37 |
| Global Warming | t $CO_2$-eq. | −21.97 | −25.37 | 21.97 |
| Marine Eutrophication | t N-eq. | 0.35 | 0.42 | −0.35 |
| Freshwater Eutrophication | t P-eq. | −0.00 | −0.01 | 0.00 |
| Fossil Resource Depletion | MJ | −551.79 | −629.00 | 551.79 |
| Human Toxicity | t 1,4-DB-eq. | −8.79 | −11.49 | 8.79 |
| Photochemical Ozone Formation | t NMVOC-eq. | 0.56 | 0.47 | −0.56 |
| Particulate Matter | t $PM_{10}$-eq. | 0.51 | 0.60 | −0.51 |
| Water Depletion | $m^3$ water | −180.87 | −194.50 | 180.87 |

(Functional unit: 1 tonne household food waste).

However, environmental damages increase for air and water quality; Slorach et al. [23] found that the environmental damage of NMVOCs, Particulate Matter Formation and Marine Eutrophication related to anaerobic digestion and incineration is larger than the potential savings which are achieved via displaced grid energy.

#### 3.3.2. Changes to the Impact of Waste Disposal Method under Each Scenario
#### Scenario 1 (20% Change in Recycling Behaviour)

Table 8 shows the environmental impacts from waste processing of collected waste, reflecting the forecast impact of movement of 3888 tonnes of food waste from residual bins for energy from waste processing into recycling caddies for anaerobic digestion.

**Table 8.** Quantity and value of change in impacts under Scenario 1: Waste Processing only (rounded).

| | Unit | Base Collected | 20% Change | Net Change | Value of Change GBP |
|---|---|---|---|---|---|
| PED Primary Energy Demand | GJ | −45,500 | −50,000 | −4000 | −102,062 |
| GWP Global Warming Potential | t $CO_2$-eq. | −730 | −800 | −110 | −5510 |
| ME Marine Eutrophication | t N-eq. | 12 | 14 | 2 | 6220 |
| FE Freshwater Eutrophication | t P-eq. | −0.12 | −0.06 | −0.07 | −106 |
| FD Fossil Depletion | GJ | −18,300 | −21,000 | −2560 | −2848 |
| HT Human Toxicity | t 1,4-DB-eq. | −290 | −400 | −100 | −7598 |
| POF Photochemical Oxidant Formation | t NMVOC-eq. | 18 | 16 | −3 | −2645 |
| PMF Particulate Matter Formation | t $PM_{10}$-eq. | 17 | 20 | 3 | 72,706 |
| WD Water Depletion | $m^3$ water | −6,000,000 | −6,500,000 | −451,000 | −30,546 |

This scenario illustrates the trade-offs which occur during a switch in waste processing method. As this scenario sees an increase in the quantity of material processed by Anaerobic Digestion (AD), which is more efficient in terms of energy generation, we see reductions in Primary Energy Demand, Global Warming Potential and Water Depletion as the environmental benefits of switching from grid electricity are realised. However, the application of digestate from AD plants has been associated with increases in air and marine water pollutants, and an increase in material to AD leads to an increase in these pollutants.

Scenario 2 (20% Reduction in All Food Waste)

Table 9 shows the environmental impacts from waste processing of collected waste, reflecting the forecast reduction of 6620 tonnes of food waste, both unavoidable and avoidable, distributed across all waste disposal methods.

**Table 9.** Quantity and value of change in impacts under Scenario 2: Waste Processing only (rounded).

| | Unit | Base Collected | 20% Change | Net Change | Value of Change GBP |
|---|---|---|---|---|---|
| PED Primary Energy Demand | GJ | −45,500 | −36,400 | 9100 | 227,300 |
| GWP Global Warming Potential | t $CO_2$-eq. | −730 | −580 | 150 | 7100 |
| ME Marine Eutrophication | t N-eq. | 12 | 9 | −2 | −6140 |
| FE Freshwater Eutrophication | t P-eq. | −0.12 | −0.10 | 0.02 | 40 |
| FD Fossil Depletion | GJ | −18,300 | −14,600 | 3700 | 4071 |
| HT Human Toxicity | t 1,4-DB-eq. | −290 | −230 | 60 | 4950 |
| POF Photochemical Oxidant Formation | t NMVOC-eq. | 18 | 15 | −4 | −3600 |
| PMF Particulate Matter Formation | t $PM_{10}$-eq. | 17 | 13 | −3 | −76,750 |
| WD Water Depletion | $m^3$ water | −6,000,000 | −4,800,000 | 1,200,000 | 81,100 |

Table 9 illustrates how, in this scenario, a 20% reduction in material leads to a corresponding drop in the benefits that waste disposal brings—less energy generation and digestate reduces beneficial impacts in the baseline state relating to primary energy demand, global warming potential and water depletion, interpreted here as a net cost. However, the reduction in food waste material does mitigate some baseline environmental impacts: air ($PM_{10}$ and NMVOCs) and water pollutants (N) are reduced.

*3.4. The Whole Food Cycle: The Combined Impact of Embodied Resource Use and Waste Disposal*

The following results relate to household food waste collected by Bristol Waste Company at the doorstep, either in residual waste bins or in recycling food caddies (69% of total food waste in Bristol; data on food disposed of via sewer or home composting are excluded).

3.4.1. Baseline Impacts for the Whole Food Cycle

Table 10 and Figure 4 show a comparison of the baseline impacts of the whole food cycle on collected food waste, summarised using the monetisation method to show the relative weight of environmental costs and benefits.

Two categories here are represented here as benefits (negative numbers, reflecting a reduction in environmental damages). Primary Energy Demand (PED) indicates a potential benefit, as energy generated from wasted food displaces grid electricity. However, PED is not included in the Tonini et al. assessment [7], and therefore our understanding of the total resource use in this category is incomplete. Therefore, it should not be assumed that the energy generated from waste disposal outweighs the energy which goes into food production. Water Depletion is the other element which appears as a net benefit. We explore possible reasons for this result in the Discussion section below.

**Table 10.** Total environmental impacts of collected food waste in Bristol (33,101 tonnes) per annum, including resource loading and waste disposal.

| | Unit | All Collected | All AFW Collected | Value (All) GBP |
|---|---|---|---|---|
| Acidification | Mol H+ × 1000 | 1000 | 700 | 283,095 |
| Ecotoxicity | CTUe × 1000 | 117,000 | 85,000 | 3844 |
| FRD Fossil Resource Depletion | GJ | 609,000 | 444,000 | 678,505 |
| GWP Global Warming Potential | t $CO_2$-eq. | 75,000 | 55,000 | 3,688,000 |
| HT Human Toxicity | t 1,4-DB-eq. | −300 | −160 | −24,700 |
| HT Human Toxicity Cancer | CTUh | 1 | 1 | 38,700 |
| N Aquatic Eutrophication Nitrogen | t N-eq. | 430 | 311 | 1,146,200 |
| P Aquatic Eutrophication Phosphorous | t P-eq. | 11 | 8 | 17,200 |
| PED Primary Energy Demand | GJ | −45,000 | −31,000 | −1,136,500 |
| PM Particulate Matter | t $PM_{10}$-eq. | 17 | 11 | 383,800 |
| PM Particulate Matter | t $PM_{2.5}$-eq. | 66 | 48 | 2,196,400 |
| POF Photochemical Ozone Formation | t NMVOC-eq. | 200 | 200 | 238,400 |
| WD Water Depletion | $m^3$ water | −6,000,000 | −4,000,000 | −396,100 |

Quantities rounded. AFW: Avoidable Food Waste.

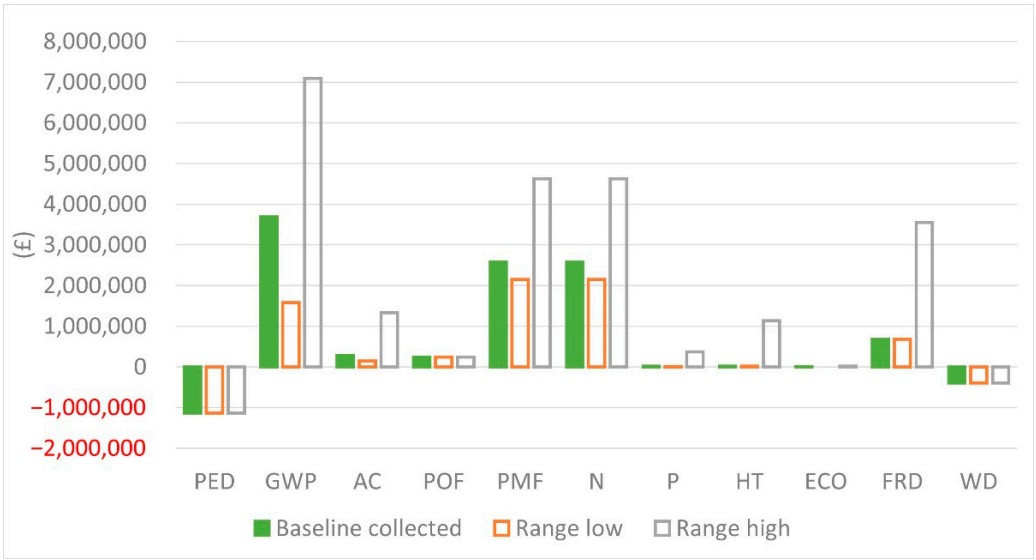

**Figure 4.** Baseline value of impacts for whole food cycle for all collected waste in Bristol (33,101 tonnes), GBP per annum. Key: PED: Primary Energy Demand; GWP: Global Warming; AC: Acidification; POF: Photochemical Ozone Formation; PMF: Particulate Matter Formation (PM2.5 and PM10 combined); N: Aquatic Eutrophication (Nitrogen), P: Aquatic Eutrophication (Phosphorous); HT: Human Toxicity (CTUh and t 1,4-DB equiv. combined); ECO: Ecotoxicity; FRD: Fossil Resource Depletion; WD: Water Depletion.

3.4.2. Comparison of Impacts for the Whole Food Cycle under Each Scenario
Scenario 1: Increased Recycling

It should be observed that in Scenario 1, the estimated changes for avoidable food waste are much larger in some cases than the "All" waste category. This relates to the original data for Bristol food waste—the proportion of avoidable to unavoidable food waste is reduced when waste is recycled. For recycled material avoidable waste comprises 58% of food, but for residual waste the avoidable component comprises 83%. Therefore, when more food is recycled, this change in ratio affects the size of the change in environmental impact.

There is no change to resource loading of food in Scenario 1, but the effects observed below in Table 11 are a combination of the change in the ratio of avoidable/unavoidable food waste, and the trade-offs between different methods of waste disposal.

**Table 11.** The effect of the whole food cycle with 20% increased recycling.

| | Unit | All Collected | All AFW Collected | Value (All) GBP | Value (AFW) GBP |
|---|---|---|---|---|---|
| PED Primary Energy Demand | GJ | −4000 | −1400 | −102,062 | −35,306 |
| GWP Global Warming Potential | t CO$_2$-eq. | −113 | −2355 | −5510 | −115,091 |
| Acidification | Mol H+ × 1000 | 0 | −29 | 0 | −8546 |
| POF Photochemical Ozone Formation | t NMVOC-eq. | −3 | −9 | −2645 | −9001 |
| PM Particulate Matter | t PM$_{10}$-eq. | 3 | 2 | 72,706 | 38,005 |
| PM Particulate Matter | t PM$_{2.5}$-eq. | 0 | −2 | 0 | −66,305 |
| N Aquatic Eutrophication Nitrogen | t N-eq. | 2 | −11 | 6220 | −30,359 |
| P Aquatic Eutrophication Phosphorous | t P-eq. | 0 | 0 | −106 | −592 |
| HT Human Toxicity Cancer | CTUh | 0 | 0 | 0 | −1169 |
| HT Human Toxicity | t 1,4-DB-eq. | −89 | −52 | −7598 | −4436 |
| Ecotoxicity | CTUe × 1000 | 0 | −4000 | 0 | −116 |
| FRD Fossil Resource Depletion | GJ | −2600 | −20,100 | −2848 | −22,425 |
| WD Water Depletion | m$^3$ water | −450,000 | −130,000 | −30,546 | −8876 |

Total net quantity and societal value of change in environmental impacts of collected food waste in Bristol (33,101 tonnes) per annum, including resource loading and waste disposal under Scenario 1. Quantities rounded. AFW: Avoidable Food Waste.

Scenario 2: Reduction in Food Waste

In Scenario 2 all impacts change proportionally, so reductions in benefits such as Primary Energy Demand and Water Depletion appear as cost increases (Table 12).

**Table 12.** The effect of the whole food cycle with 20% reduced food waste.

| | Unit | All Collected | All AFW Collected | Value (All) GBP | Value (AFW) GBP |
|---|---|---|---|---|---|
| PED Primary Energy Demand | GJ | 9000 | 6000 | 227,290 | 154,400 |
| GWP Global Warming Potential | tCO$_2$-eq. | −15,000 | −11,000 | −737,600 | −536,775 |
| Acidification | Mol H+ × 1000 | −192 | −140 | −56,619 | −41,159 |
| POF Photochemical Ozone Formation | t NMVOC-eq. | −48 | −35 | −47,680 | −34,940 |
| PM Particulate Matter | t PM$_{10}$-eq. | −3 | −2 | −76,750 | −48,080 |
| PM Particulate Matter | t PM$_{2.5}$-eq. | −13 | −10 | −439,285 | −319,330 |
| N Aquatic Eutrophication Nitrogen | t N-eq. | −86 | −62 | −229,240 | −165,990 |
| P Aquatic Eutrophication Phosphorous | t P-eq. | −2 | −1.5 | −3440 | −2515 |
| HT Human Toxicity Cancer | CTUh | −0.2 | −0.2 | −7742 | −5628 |
| HT Human Toxicity | t 1,4-DB-eq. | 58 | 33 | 4945 | 2789 |
| Ecotoxicity | CTUe × 1000 | −23,000 | −17,000 | −769 | −559 |
| FRD Fossil Resource Depletion | GJ | −122,000 | −89,000 | −135,701 | −98,948 |
| WD Water Depletion | m$^3$ water | 1,170,000 | 803,000 | 79,229 | 54,355 |

Total net quantity and societal value of change in environmental impacts of collected food waste in Bristol (33,101 tonnes) per annum, including resource loading and waste disposal under Scenario 2. Quantities rounded. AFW: Avoidable Food Waste.

We combine data from Tables 10–12 in Figure 5 below. Here, we show the baseline effect of the whole food cycle for collected waste, and the net change under each scenario, in order to illustrate and compare the relative scale of change.

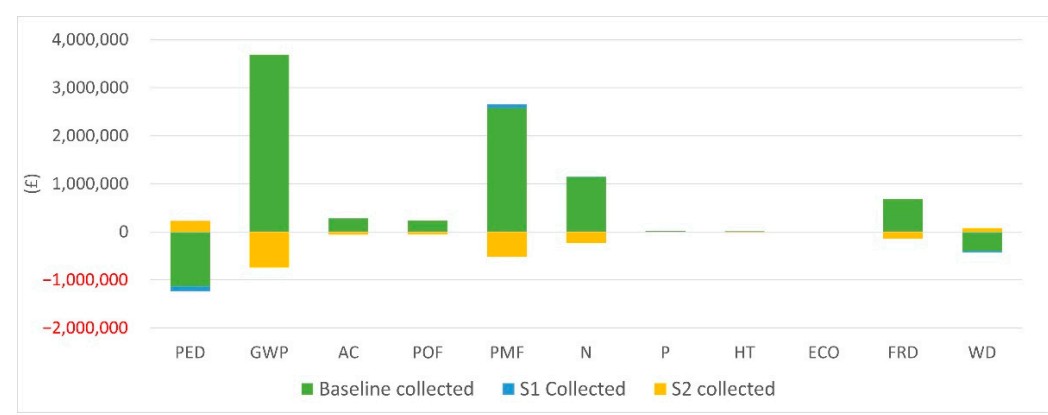

**Figure 5.** Comparison of societal value of environmental impacts at baseline and net effect of 20% change under each scenario. (All Bristol collected waste (33,101 tonnes), GBP per annum). Key: PED: Primary Energy Demand; GWP: Global Warming; AC: Acidification; POF: Photochemical Ozone Formation; PMF: Particulate Matter Formation (PM2.5 and PM10 combined); N: Aquatic Eutrophication

(Nitrogen), P: Aquatic Eutrophication (Phosphorous); HT: Human Toxicity (CTUh and t 1,4-DB equiv. combined); ECO: Ecotoxicity; FRD: Fossil Resource Depletion; WD: Water Depletion.

For example, the value of the impact of Global Warming Potential (GWP) for all collected waste is estimated at GBP 3.7 million (Table 10). Under Scenario 1, we estimate that a 20% increase in recycling may lead to a small reduction in GWP of GBP 5500 (Table 11). Under Scenario 2, we estimate that a 20% reduction in food waste at source would lead to a reduction in GWP of GBP 738,000 (Table 12).

For Particulate Matter Formation (PMF) (PM2.5 and PM10 combined), the value of the impact of this at baseline is estimated as a total of GBP 2.6 million (Table 10). The estimated effect of increased recycling is an increase in pollution valued at GBP 73,000 (Table 11). The estimated effect of reduced food waste at source is a reduction in PMF of GBP 516,000 (Table 12).

*3.5. The Value of Food Waste to Households*

In all of the impacts we have outlined in the section above, we have explored the societal cost, which is inherently an externality for the households generating this waste.

The value of food waste in terms of the market cost of the avoidable element of food is used by WRAP in their campaign for Love Food Hate Waste [11]. It is an easily understandable metric for the scale and impact of food waste to individual households.

Estimates for the value of food waste vary, depending on how metrics are aggregated between the individual, family, household or national level, and whether quantities are given per kilo, or per average quantities of waste per week or month or year.

We include value of food waste to households only as a comparison with the environmental impact of food waste. We recognise that there are more complexities relating to what happens when households reduce consumption—behaviour changes mean there is not automatic saving of household income, and choices about what to do with any surplus affect the environmental gains or losses [12]. For example, a Danish study exploring the life cycle costs of food waste management found that in some instances, substitute goods bought with surplus income from food waste savings could have a larger environmental impact than the original goods [40].

We use a value per kilo of edible food waste based on WRAP (2020) [2], using their estimate for the average value per month of edible food waste for each household in the UK, which is estimated at GBP 2.86 per kilo. This is at the lower end of the range which WRAP applies; values range from GBP 2.86 to GBP 3.10 per kilo depending on how the quantities are aggregated.

We find that on average, Bristol households waste around 240 kg per year in total and, of this, 172 kg is avoidable. This equates to around 3.31 kg avoidable food waste per week per household. We estimate that this has a value of around GBP 41 per month, or GBP 490 per year per Bristol household.

The total avoidable food waste per household is just over the 2020 national WRAP estimate of 165 kg avoidable waste per household per year. The Bristol rates of average food waste per household per week are slightly lower than the UK average of 3.4 kg per week per household, with an annual value of GBP 500 per UK household [2] (Figure 6).

We also show below what might happen to average household quantities of avoidable food waste per household under each scenario. Figure 7 (Scenario 1) shows that the overall quantity of food waste does not change, and therefore there is no financial benefit to the household. Recycling rates improve from 30% to 38%. Figure 8 (Scenario 2) shows that although there is no change to recycling rates, the overall quantity of food wasted reduces by 20%.

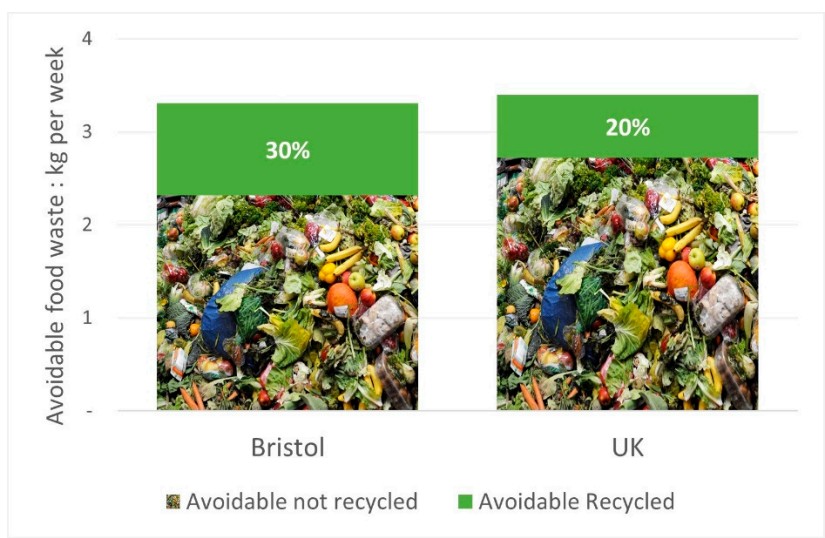

**Figure 6.** Average weekly avoidable food waste in Bristol compared to the UK.

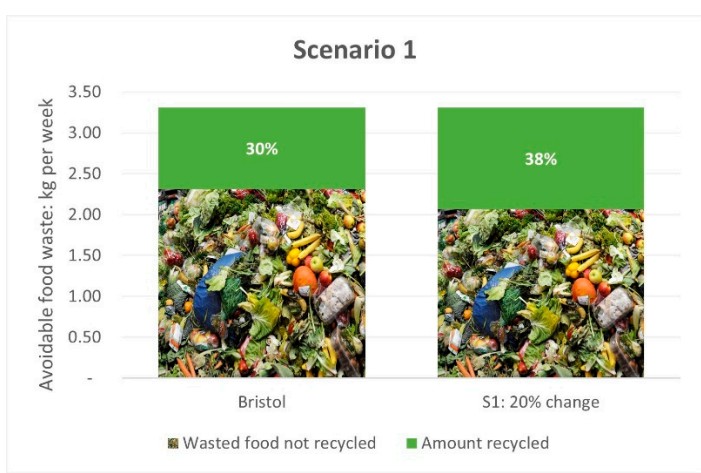

**Figure 7.** Changes to average weekly household AFW with improved recycling at 20%.

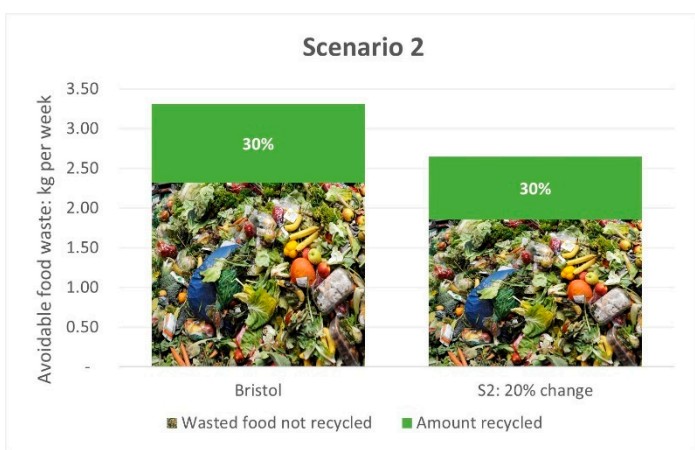

**Figure 8.** Changes to weekly household AFW with reduced food consumption at 20%.

## 4. Discussion

### 4.1. Limitations

There are some specific limitations related to our methodology:

We focus on food waste collected by Bristol Waste Company via domestic collections from households, so this may exclude a small number of residents of Bristol, for example, those who live in flats and other forms of accommodation which are not serviced by Bristol Waste Company.

Estimations of environmental impacts, and any benefits from displacement of alternative products such as grid energy, are based on a point in time. This study does not have the scope to model how these interactions might change if, for example, UK grid energy becomes supported by more renewable energy generation. However, recent energy trends indicate that the mixture of renewables to fossil fuels for grid energy generation has been subject to some variation [41]. Energy generated from biogas may not perform as well when compared to other renewable methods [42].

Using economic valuation techniques alongside Life Cycle Analyses (LCAs) allows us to compare the relative scale of each environmental effect and is a useful method to assess potential benefits and costs of policy interventions. However, the methods used to identify directly attributable environmental impacts mean that valuations of externalities appear extraordinarily low, if we compare these to market values. Some life cycle analyses have been shown to underestimate environmental effects due to truncation errors [15]. Our results are likely to be underestimates because the system boundaries imposed in LCA methods may not include all indirect effects attributable to food waste [43]. Environmental prices are also subject to large uncertainties due to methods of calculation; for example, the 2015 midpoint price per kilo $CO_2$ equivalent ranges from EUR 0.0218 to EUR 0.0944 and is expected to increase to between EUR 0.073 and EUR 0.315 by 2050 [37].

For example, the value of Bristol's avoidable food waste is estimated in the region of GBP 100 million per year. The abatement cost of greenhouse gas emissions relating to this waste is estimated to be in the region of GBP 5.3 million per year. The two metrics are, of course, not comparable, but without a hybrid approach which brings together market and non-market values cost–benefit analysis might lead to conclusions that waste is too costly to prevent. Future research in this area would be useful in order to explore this issue. However, the benefit of using environmental prices with LCA remains as a tool to apply weighting factors to different environmental outcomes [37].

We assume that local waste processing practices in Bristol have the same environmental footprint as set out in Slorach et al. [22], which took evidence from UK waste processing plants. However, it has been shown that significant variations in environmental impacts may occur with individual waste processing techniques. For example, alternative methods for disposal of digestate from anaerobic digestion plants may have significant implications for the amount of particulate matter released into the air [44,45].

In the estimates for our scenarios, we show the maximum potential change in order to compare the relative effects of interventions, but factors such as choice of alternative foods, buying behaviour, etc. mean that this maximum in terms of environmental benefit or freed effective income may not be reached, for example, if people select more meat instead of, for example, buying and wasting potatoes. Saleemdeeb et al. finds that changes in buying behaviour and food choices means that there could be a significant rebound effect with a food waste reduction intervention, potentially reducing the environmental benefit of reduced food waste by up to 60% [12].

Households are the final link in a whole chain of food production and supply, and the scope of this study does not address the choice architecture which determines a household's disposition to waste food, or how this may be altered by changes at other points in the food cycle, for example, changes in how supermarkets and food producers influence the quality, storage and availability of food. Behaviour leading to food waste is extremely complex, and we need more understanding about where food waste might be displaced in the food cycle if households reduce consumption.

### 4.2. Key Points for Environmental Improvement and Policy

Bristol is a useful example for demonstrating why national-level data are no substitute for local-level analysis. Bristol is unusual compared to the rest of the UK in that not every local authority offers a free food waste recycling collection service at the doorstep, and waste disposal methods vary widely in the UK, with some areas more reliant on landfill, which has a much larger environmental impact than the methods used in Bristol [23,28]. Our methodology allows the policy maker to adjust national-level estimation to account for significant local variations in waste management practices.

Our findings illustrate the wide difference between reducing consumption and recycling material in a powerful reminder of the waste hierarchy [46]. In accordance with the waste hierarchy, we find that, compared to incineration or landfill, anaerobic digestion is found to have the most positive environmental outcomes of those waste disposal methods compared, although some methods can increase particulate matter pollution. The optimisation of food waste reduction strategies should take into account the life cycle of the disposal method, and what resources any outputs replace.

Twenty-eight percent of all wasted food in Bristol is unavoidable or not edible, and this waste should be disposed of in the most efficient way possible. The environmental benefits of improvements in recycling seen here would suggest that campaigns such as Bristol's "Slim my Waste, Feed my Face" initiative are valuable and should be extended [35].

### 4.3. Comparison of Results with Other Studies

Our findings agree with those of many other studies which conclude that food waste prevention may lead to substantial reductions in environmental impacts, including climate change and depletion of finite resources. However, there are uncertainties in the magnitude of effect.

Chapagain and James estimated that the total water footprint of all household food waste in the UK is 6262 million $m^3$ per year [14]. Total household food waste in this year was 7.2 million tonnes [47]. We estimate that this would be the equivalent of 869.7 $m^3$ water per tonne food waste. The 2011 study combined internal and external water footprints. Using the same method, we estimate the internal footprint to be 251 $m^3$ per tonne and 618 $m^3$ per tonne for the external footprint.

The amount of water detailed in Tonini et al. [7] (4.17 $m^3$ per tonne) is potentially 200 times less than the Chapagain and James study [14]. Chapagain and James calculate the water footprint based on direct water use and indirect water use; including water used in the supply chains throughout the food cycle. However, Tonini et al. use a database which only calculates direct Water Depletion, so water where it forms an ingredient or element of food processing, for example. It does not include indirect water use in energy generation or vehicle movements, for example.

Therefore, the Tonini et al. estimate for Water Depletion in the Resource Loading element of our calculations should be seen as the direct water component of food waste only, rather than the wider water footprint of all food production.

This uncertainty over definitions of Water Depletion may account for the anomaly where we find only 200,000 $m^3$ water is embodied into food waste from production, whereas we estimate 6 million $m^3$ water are saved due to waste disposal methods. We tested the same results using the definition of water depletion for resources loaded into food waste using the Chapagain and James study [14] and found that rather than the relatively small value of 200,000 $m^3$ water, the revised resource burden for all food waste could be as high as 41 million $m^3$ water. Under Scenario 2, this would mean that rather than resulting in a net increase in cost of GBP 80,000 per annum, this scenario could lead to savings of GBP 309,000 per annum.

Chapagain and James also estimate the carbon footprint of food waste to be higher than our estimate; around 3.8 tonnes of $CO_2$ equivalent per tonne of food waste, compared to Tonini et al.'s 2.5 tonnes, and our estimate of 2.28 tonnes of $CO_2$ per tonne food waste.

Saleemdeeb et al. [12] focus on Greenhouse Gas (GHG) emissions from household food waste in the UK. They use a hybrid life cycle assessment model coupled with a highly disaggregated input output analysis to capture environmental impacts across the global food supply chain for the whole life cycle of food from production to waste processing. The study only considers anaerobic digestion, but estimates the mitigating impact of AD on GWP to be much higher; $-89$ kg $CO_2$-equiv compared to the value given in Slorach et al. as $-39$ kg $CO_2$-equiv [23].

## 5. Conclusions

The nexus between food, energy and water allows us a lens to explore the implications of food waste in Bristol. Energy and water are used throughout the food cycle in the production and supply of food as resources in themselves, leading to depletion of finite resources such as fossil fuels, as well as the degradation of water and air quality through pollutants.

Bristol's annual burden of 48,000 tonnes of food waste per year could contribute around 110,000 tonnes $CO_2$ equivalent towards climate change. The societal cost of this damage, excluding effects related to waste disposal, is valued at around GBP 5.3 million per year (range GBP 2.3–10.4 million). This is based on 2015 prices; the abatement cost of carbon is expected to increase almost fourfold by 2050 [37].

The disposal of Bristol's collected food waste supports the principle of a circular economy in which resources such as water and energy are saved in order to go back into food production. The positive benefits of energy generation and fertilisers from waste, and their displacement of grid energy and traditional fertilisers, are important factors in any relative benefits derived from recycling material. If we only explore the relative efficiencies of waste disposal methods, both recycling and reduction scenarios show trade-offs between positive and negative effects at the 20% reduction level.

This may not be the case in the long term. The value of displacement of grid energy may significantly reduce if the mix of fuel sources changes towards more renewables and away from fossil fuel-derived energy [41]. The benefits of sustainable energy derived from anaerobic digestion and incineration could become outweighed by the net environmental costs.

If we include the resources which are loaded into food before the point of waste, any benefits from recycling mitigate a relatively small number of environmental damages and may increase damage to air and water quality. Given the extent of food waste in Bristol, our findings suggest that interventions towards minimising food waste at source could lead to significant reductions in potential environmental damages. Any food waste policies should holistically consider the impact of efficiencies across the whole food cycle. Furthermore, the connection between efficiencies in the food supply and consumption system, and inequalities of access to food supply, needs to be strengthened.

We have noted the limitations of the LCA-based methodology applied in this paper, and some potential benefits of hybrid approaches to quantification of environmental impacts of food waste. This micro, non-market valuation was developed in the same ULL project, and in concert with a linked integrated approach to macro-level valuation, which sought to stress-test Bristol's food waste reduction targets using macro-economic valuation and scenario planning in order to understand and overcome potential barriers [48]. Future academic research into how environmental prices can be incorporated into hybrid approaches could help to explore how interventions to reduce food waste might affect both the food economy and its environmental footprint, providing further insights to support policy appraisal at the municipal level.

**Author Contributions:** Conceptualisation—E.E., A.H., D.B.; Methodology—E.E.; Data Validation—A.H., G.F., S.H.; Formal Analysis—E.E. and A.D.L.; Investigation—E.E. and A.D.L.; Data Curation—E.E.; Writing—Original Draft Preparation—E.E.; Writing—Review and Editing—E.E., A.H., D.B., A.D.L., S.H., G.F.; Supervision—A.H.; Funding Acquisition—A.H. and D.B. All authors have read and agreed to the published version of the manuscript.

**Funding:** The WASTE FEW ULL project was funded by JPI Europe and the Belmont Forum's Sustainable Urbanization Global Initiative. The Bristol ULL was funded by the Economic and Social Research Council (ESRC) grant number ES/S002243/1, and AHRC, Innovate UK, RCN and NSF.

**Institutional Review Board Statement:** Not applicable.

**Informed Consent Statement:** Not applicable.

**Data Availability Statement:** Data on quantities of collected waste are available from www.wastedataflow.org (accessed on 22 March 2022). Restrictions apply to the availability of data relating to composition of avoidable and unavoidable food waste in Bristol. Data were obtained from Bristol Waste Company and are available from the authors with the permission of Bristol Waste Company and Resource Futures.

**Acknowledgments:** This research was developed in consultation with Bristol stakeholders, including Bristol Waste Company, Wessex Water, GENeco, Bristol Food Network, Resource Futures, the Centre for Sustainable Energy and the Schumacher Institute. This enabled us to understand and map the local resource flows and refine the model. We acknowledge their important contribution to this work and thank them for their valuable input and support.

**Conflicts of Interest:** The authors declare no conflict of interest. The funders had no role in the design of the study; in the collection, analyses or interpretation of data; in the writing of the manuscript, or in the decision to publish the results.

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
