# Peer review of "What Are the Environmental Benefits and Costs of Reducing Food Waste? Bristol as a Case Study in the WASTE FEW Urban Living Lab Project"

_sustainability, doi:10.3390/su14095573_

Round 1

Reviewer 1 Report

Very interesting approach, aiming to set a model, test and quantify the environmental impacts of changes in food waste recycling behaviours. Having in mind that, as Authors stated, about 30% of food production is wasted in global food production shortages, the significance of the presented analysis should be highlighted. The specific contribution is seen in approach to quantify the value of the environmental impact of changes to resource availability and pollution as well as a results of reducing the food waste. Environmental prices (as consequence) as such are hard to be foreseen but the presented approach uses the values for environmental prices that are derived from studies that use a Life Cycle Assessment approach, together with values for pollutants damage costs, ecosystem impacts and resource availability. Including the socio-environmental impacts from household food waste certainly would affect to changes to food waste behaviour. The Authors also include key points for environmental improvement and policy which I find additional contribution to the presented research.

The motivation of conducted analysis is quite strong, clear and well presented. The Introduction part is well structured, but it should be extended with similar studies conducted in recent years, the scientific ones. Including the report results is inevitable to conduct the reliable data, but the scientific background of the research is quite weak.

It seems that the paper was prepared according to different journals’ instructions for authors as the mentioned references include the style mixture, included the name and the year, and numerically mentioned references. It should be corrected, as it is quite hard to read and understand the data sources. The structure of the paper should be given in the end of the Introduction section.

The introduction should be revised. The comparison of existing methodologies should be more clearly presented, and justification of presented methodology should be provided. And please emphasize the novelty and academic significance of your work.

The strengths and weaknesses of applied approach are needed to be highlighted and included in the study.

I suggest that the Authors should addressed future research guidelines and provided them in conclusion.

Reviewer 2 Report

The authors focus on “What are the environmental benefits and costs of reducing food waste?”.

The aim of this paper is to model, test and quantify the environmental impacts of changes in food waste recycling behaviours in the area of Bristol.

The authors using detailed research, accurate data and rational analysis, answered the two questions:

  1. What are the non-market and socio-environmental benefits of reduced food waste along the food/waste cycle through increased food waste recycling?
  2. What reductions in energy and other resource usage in food production/transport and waste disposal might be gained from reducing food waste by 20%?

   In conclusion, any food waste policies should consider holistically the impact of efficiencies across the whole food cycle.

Author Response

Point 1:  Are the conclusions thoroughly supported by the results presented in the article or referenced in secondary literature? (This item has been marked as “Can be improved”)

Thank you very much for your helpful comments. We have revised our conclusion section to ensure that all conclusions are more clearly referenced.

Point 2: The authors focus on “What are the environmental benefits and costs of reducing food waste?”.

The aim of this paper is to model, test and quantify the environmental impacts of changes in food waste recycling behaviours in the area of Bristol.

The authors using detailed research, accurate data and rational analysis, answered the two questions:

  1. What are the non-market and socio-environmental benefits of reduced food waste along the food/waste cycle through increased food waste recycling?
  2. What reductions in energy and other resource usage in food production/transport and waste disposal might be gained from reducing food waste by 20%?

In conclusion, any food waste policies should consider holistically the impact of efficiencies across the whole food cycle.

Thank you very much for these comments.  We have updated our conclusion to ensure that this point is emphasised.

Reviewer 3 Report

The manuscript entitled “What are the environmental benefits and costs of reducing food waste? Bristol as a case study in the WASTE FEW Urban Living Lab Project” is an intriguing study. The authors estimate the environmental costs and benefits of reducing food waste in Bristol, UK. The paper is well-written, well-structured, and the results are valuable. This study has the potential to attract a wider audience. The strength of this paper lies in the detailed and comprehensive discussion of research methods and variables used in this study. My comments and suggestions are minimal:

The introduction should include more literature and a description of the research gap.

Additionally, the authors should discuss the paradoxical debate surrounding food waste. See the following:

Pinstrup-Andersen, Per, Vincent Gitz, and Alexandre Meybeck. "Food losses and waste and the debate on food and nutrition security." In Routledge handbook of food and nutrition security, pp. 187-202. Routledge, 2016.

In the two scenarios discussed by the authors, is there a particular reason to assume a 20% increase and decrease?

All limitations of the study should be acknowledged in the limitations section (including those mentioned in other sections). 

Round 2

Reviewer 1 Report

Dear Authors,

The paper has been significantly changed according to the given remarks. Thank You for such interesting analysis.